REVIEW-SYMPOSIUM

# Human retinal dark adaptation tracked *in vivo* with the electroretinogram: insights into processes underlying recovery of cone- and rod-mediated vision

Xiaofan Jiang[1,2,3,4] and Omar A. Mahroo[1,2,3,4,5] (iD)

[1] *Institute of Ophthalmology, University College London, London, UK*
[2] *Retinal and Genetics Services, Moorfields Eye Hospital, London, UK*
[3] *Section of Ophthalmology, King's College London, London, UK*
[4] *Department of Twin Research and Genetic Epidemiology, King's College London, St Thomas' Hospital Campus, London, UK*
[5] *Physiology, Development and Neuroscience, University of Cambridge, Cambridge, UK*

Edited by: Laura Bennet & William Taylor

The peer review history is available in the Supporting Information section of this article (https://doi.org/10.1113/JP283105#support-information-section).

**Xiaofan Jiang** is a PhD student at University College London, having previously completed an MSc with distinction at King's College London. Her research is supervised by Omar Mahroo, and explores the use of electroretinography to probe aspects of human retinal function in health and disease *in vivo*. She has presented her findings at several international meetings. **Omar Mahroo** is Professor of Retinal Neuroscience at University College London, and Consultant Ophthalmologist at Moorfields Eye Hospital and St Thomas' Hospital in London. He completed his medical degree and PhD at the University of Cambridge. His PhD (2001–2004) at Cambridge, and post-doctoral work at the Australian National University, were supervised by Trevor Lamb FRS. The resulting papers were published in the *Journal of Physiology*. He completed his ophthalmology training in London, including an Academic Clinical Lectureship at King's College London, and a retinal fellowship at Moorfields Eye Hospital.

This review was presented at the Physiology 2021 symposium 'Photoreceptors in Health and Monogenic Disease: Advances in Understanding Physiology and Treating Pathophysiology' on 16 July 2021, organised by Dr Omar Mahroo, UCL Institute of Ophthalmology, UK.

**Abstract**   The substantial time taken for regaining visual sensitivity (dark adaptation) following bleaching exposures has been investigated for over a century. Psychophysical studies yielded the classic biphasic curve representing recovery of cone-driven and rod-driven vision. The electro-retinogram (ERG) permits direct assessment of recovery at the level of the retina (photoreceptors, bipolar cells), with the first report over 70 years ago. Over the last two decades, ERG studies of dark adaptation have generated insights into underlying physiological processes. After large bleaches, rod photoreceptor circulating current, estimated from the rod-isolated bright-flash ERG a-wave, takes 30 min to recover, indicating that products of bleaching, thought to be free opsin (unbound to 11-*cis*-retinal), continue to activate phototransduction, shutting off rod circulating current. In contrast, cone current, assessed with cone-driven bright-flash ERG a-waves, recovers within 100 ms following similar exposures, suggesting that free opsin is less able to shut off cone current. The cone-driven dim-flash a-wave can be used to track recovery of cone photopigment, showing regeneration is 'rate-limited' rather than first order. Recoveries of the dim-flash ERG b-wave are consistent also with rate-limited rod photopigment regeneration (where free opsin, desensitising the visual system as an 'equivalent background', is removed by rate-limited delivery of 11-*cis*-retinal). These findings agree with psychophysical and retinal densitometry studies, although there are unexplained points of divergence. Post-bleach ERG recovery has been explored in age-related macular degeneration and in trials of visual cycle inhibitors for retinal diseases. ERG tracking of dark adaptation may prove useful in future clinical contexts.

(Received 15 March 2022; accepted after revision 4 May 2022; first published online 25 May 2022)

**Corresponding author** O. A. Mahroo: Institute of Ophthalmology, University College London, Bath Street, London, EC1V 9EL UK.     Email: o.mahroo@ucl.ac.uk

**Abstract figure legend** The prolonged time course of recovery of human visual sensitivity (dark adaptation) following bright light exposure has been the subject of study for over one and a half centuries. The upper panels represent (illustratively rather than accurately) recovery in visual sensitivity over time following a bright light exposure ('bleach'). Various techniques have been employed to investigate this, ranging from psychophysical to electrophysiological to densitometric. Psychophysical assessments were earliest and more accessible, assessing higher levels of the visual system (conscious perception). Densitometric assessments came later and allowed direct measurement of photopigment levels. Electrophysiological measurements are somewhere in between. The years shown represent approximately the earliest year in which a deliberate quantitative investigation of recovery following bleaching was performed or published. This review mainly considers those ERG studies conducted in the last 20–25 years, and the range of insights gained from these studies (summarised in the right-hand box). The lowest row of panels gives an example: recovery of the dim-flash ERG b-wave amplitude with time in the dark after a bright light exposure (traces are schematic responses expected in a healthy human subject). The panels in this row show the response to the same strength of flash (ERG amplitude plotted against time after flash in each panel) delivered at a progressively later time in the dark following extinction of a bright background (in this case a steady-state exposure to approximately 3000 scotopic trolands). Abbreviations: ERG, electroretinogram; ERP, early receptor potential; VEP, visual-evoked potential.

## Introduction

The visual system adapts by appropriately adjusting sensitivity over a remarkable range of background intensities spanning more than nine log units. Much of this adaptation occurs at the level of the retina, with many of the underlying processes occurring within the first order neurons of the visual system, the photo-receptors themselves. Whilst light adaptation is rapid, dark adaptation, the recovery of sensitivity following an exposure that has bleached a significant proportion of photopigment, is slower, and can take over 30 min for full recovery of scotopic sensitivity following a near total bleach. The kinetics of this recovery can reveal much about the cellular mechanisms in healthy individuals and potentially in patients with retinal disease.

This review will first discuss briefly some of the methods used to track visual system recovery in humans following bleaching exposures, before discussing in more detail approaches that have been taken using the electro-retinogram (ERG), focusing on those conducted over the last 20–25 years. The ERG represents the summed electrical response of the retina to light stimuli. Particular stimulus protocols can be devised to selectively investigate recovery of rod- or cone-driven responses, and analysis of particular waveform parameters may shed light on responses of photoreceptors or bipolar cells. As the various

approaches are considered, some of the insights yielded into retinal dark adaptation mechanisms will also be highlighted. The final concluding section will include consideration of potential future clinical applicability.

## Approaches to monitoring human dark adaptation

**Psychophysical methods have been used for over a century and revealed the classic biphasic recovery following cessation of bright light exposures.** The decline in threshold intensity at which an individual consciously perceives stimuli following bright light exposures was measured over a century and a half ago by Aubert in 1865. Later studies showed clearly that recovery was biphasic (Hecht et al., 1937; Kohlrausch, 1922), though the non-monotonic nature of recovery has been identified in the earlier findings of Aubert (Dodt, 1983). Following a large bleach, an initial rapid decline in threshold occurs that reaches a plateau after around 5 min; a slower decline proceeds subsequently that can take over 30 min to reach a plateau. The two components correspond to the recovery in cone-mediated and rod-mediated sensitivity, respectively.

Figure 1A depicts typical recoveries following a range of bleaches, showing the classic biphasic recovery particularly evident following the largest bleaches. The bleach level depends on the overall strength and duration of the bleaching exposure, with recovery kinetics dependent on the bleach level (regardless of whether the bleach arose from a steady exposure of a given intensity or a brief exposure to higher intensity light). Recoveries can vary according to location of the test stimulus on the retina. The spectral composition of the bleach and of test stimuli will also determine the extent to which rod- or cone-mediated sensitivity is being investigated.

Figure 1A also shows that the form of the rod-mediated portion of recovery has more than one component. As the y-axis plots threshold elevation on a logarithmic scale, an exponential recovery would manifest as a straight line, but the recoveries do not show a uniform gradient. However, as noted by Lamb (1981), and later discussed in detail by Lamb and Pugh (2004, 2006), the curves each do show a region that appears to have a uniform slope across multiple recoveries (highlighted by the orange lines in Fig. 1B). This component of recovery was termed 'S2' (Lamb, 1981) and has a slope of approximately 0.24 log units per minute. Lamb (1981) suggested that this may relate to removal of a specific photoproduct of bleaching. Interestingly the time taken to reach a particular threshold intensity in this region (example denoted by the horizontal red line in Fig. 1B) was linearly related to the fractional bleach (above a certain bleach level), which suggested that removal of this substance was not first order, but proceeded at an initial constant linear rate, which Lamb

termed 'rate limiting' behaviour. (The possible identity of the main photoproduct as free opsin, and the basis of the rate limit, will be discussed in later sections of this review.) The later, slower component of recovery (after the S2 component) was termed S3 by Lamb (1981).

**Retinal densitometry permitted direct *in vivo* quantification of photopigment regeneration.** A key requirement for restoring retinal sensitivity following

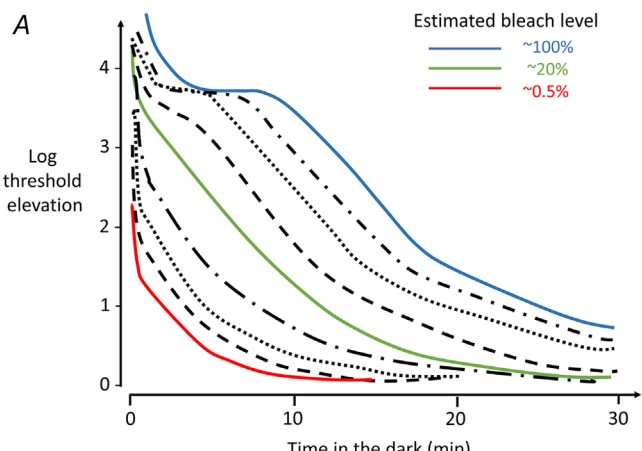

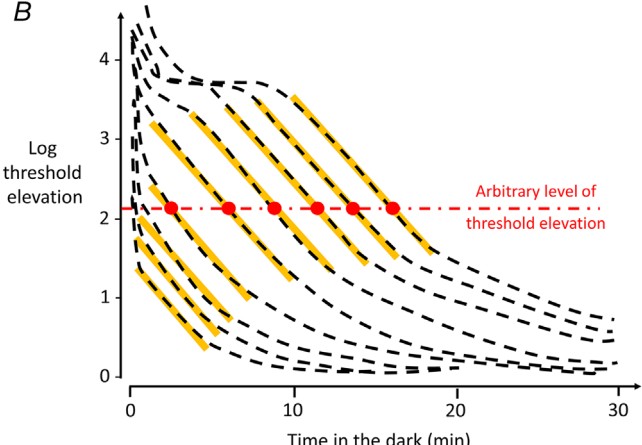

**Figure 1. Psychophysical recovery of sensitivity following a range of bleaches from a human subject**
*A*, recoveries with some lines colour-coded to give indication of estimated bleach level prior to recovery. *B*, dashed lines plot the same recoveries as in *A*, but with orange lines highlighting a region with a relatively similar slope in all of the recoveries of approximately 0.24 log units per minute, as identified by Lamb (1981) and termed the 'S2' component. The horizontal dash-dotted red line denotes an arbitrary level of threshold elevation within this region; the red circles denote the points at which recoveries cross this line. The time taken to such a level of threshold elevation (denoted by the *x*-axis value for each red circle) was shown by Lamb (1981) to be linearly related to the initial bleach level, for large bleaches, suggesting removal of a bleaching photoproduct at a constant linear rate. Redrawn (as recovery curves) from the data points of Pugh (1975), adapted from Lamb & Pugh (2004, 2006).

bleaching exposures is regeneration of photopigment. Rhodopsin comprises a vitamin A-derived chromophore, 11-*cis*-retinal, bound to a protein moiety, opsin. A photon of light isomerises 11-*cis*-retinal to all-*trans*-retinal. This activates rhodopsin, initiating the phototransduction cascade which culminates in closure of rod outer segment membrane cation channels, leading to hyperpolarisation of the photoreceptor membrane and reduction in glutamate release at the photoreceptor to bipolar cell synapse. (The ionic current that flows into the photo-receptor outer segment in the dark will be referred to as the 'circulating current' in this review, consistent with terminology in previous studies, for example Thomas & Lamb, 1999.)

The all-*trans*-retinal is released from opsin and is converted to all-*trans*-retinol (possibly prior to release). All-*trans*-retinol is transferred to the retinal pigment epithelium (RPE) where it is converted back to 11-*cis*-retinal. The 11-*cis*-retinal returns to the rod outer segment, and combines with opsin to form rhodopsin. This is termed the retinoid cycle (also called the visual cycle), and is shown in Fig. 2. In cone photo-receptor outer segments, 11-*cis*-retinal is bound to cone opsin, and similar processes occur. An additional visual cycle was identified 20 years ago (Mata et al., 2002), whereby all-*trans*-retinol can be converted to 11-*cis*-retinol within Müller cells. 11-*cis*-Retinol can be converted to 11-*cis*-retinal within cones, but not rods, and hence this has been termed a 'cone-specific' visual cycle, allowing cone outer segments access to a separate pool of chromophore, not available to rods. The retinal G-protein-coupled receptor (RGR, not shown in Fig. 2) provides further contributions to retinoid recycling (Morshedian et al., 2019; Radu et al., 2008). RGR is expressed in the RPE and in Müller cells, and mediates light-driven processes, likely to be of less relevance to recovery in the dark, which is the focus of this review.

Over six decades ago, Rushton & Henry (1955) used the technique of retinal densitometry to directly track human photopigment regeneration *in vivo*. Shining light of selected wavelengths into the eye, they quantified densities of reflected light, both during and after bleaching exposures. They showed that it took over 30 min for complete rhodopsin regeneration. Although this time course is similar to recovery of scotopic sensitivity, they noted that even when rhodopsin was 50% regenerated, the psychophysical threshold elevation is orders of magnitude higher, and so photopigment depletion by itself could not explain the large reduction in psychophysical sensitivity.

**Electrophysiology can be used to track recovery of the visual system at different levels and showed that the reduction in sensitivity is detectable in the retina.** Studies describing ERG recordings *in vivo* were published in the 19th century, and the later experiments of Granit (1933) in the decerebrate cat were important in describing separable components in the ERG, though he did not specifically investigate dark adaptation kinetics. Electrophysiology was later used specifically to explore recovery of visual system responses following bleaching exposures, with the earliest studies preceding the first retinal densitometry studies, and demonstrating that the reduction in visual system scotopic sensitivity was clearly detectable at the level of the retina. Adrian (1945) showed recovery of the ERG in human sub-jects with time in the dark, and a few years later Karpe and Tansley (1948) published their investigations of recovery of the ERG in human subjects following bleaching exposures. These studies (as well as the earlier studies of Granit) were published in *The Journal of Physiology*. Numerous investigations were published over subsequent decades (Alpern & Faris, 1954; Armington, 1959; Algvere, 1967; Auerbach, 1967; Brunette, 1969; Berson & Goldstein, 1970; Dowling, 1960; Elenius &

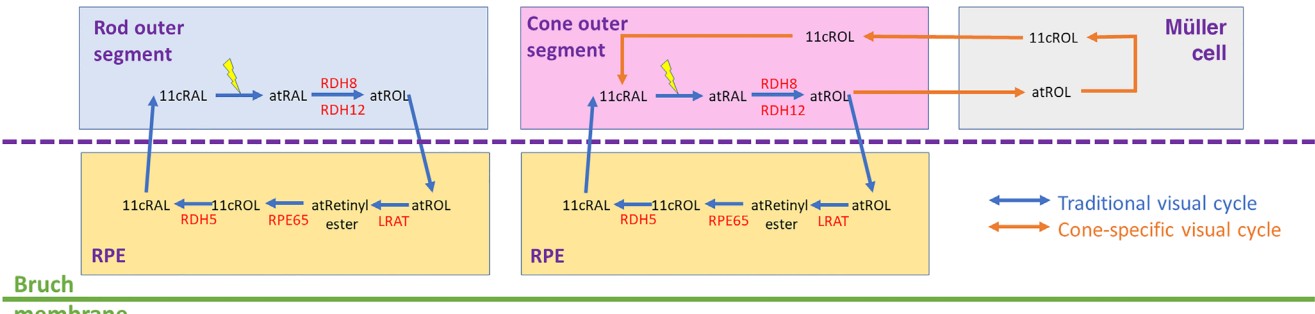

**Figure 2. Visual cycle for regeneration of 11-*cis*-retinal**
The 'traditional' visual cycle involves the retinal pigment epithelium (RPE), providing the chromophore to both rod and cone outer segments. A more recently described cycle involving Müller cells, separate from the RPE, provides 11-*cis*-retinol, which can be converted in cones, but not in rods, to 11-*cis*-retinal. 11cRAL, 11-*cis*-retinal; 11cROL, 11-*cis*-retinol; atRAL, all-*trans*-retinal; atRetinyl, all-*trans*-retinyl; atROL, all-*trans*-retinol; LRAT, lecithin retinol acyltransferase; RDH, retinol dehydrogenase; RPE65, retinal pigment epithelium-specific 65 kDa protein.

Ahlas, 1961; Granda & Biersdorf, 1966; Kawabata, 1963; Schweitzer & Troelstra, 1963; Schweitzer & Troelstra, 1964). The cortical visual evoked potential (VEP), which can be recorded non-invasively with scalp electrodes placed over the visual cortex, has also been recorded during dark adaptation, as a more objective alternative to psychophysical tests (Airas & Petersen, 1985; Fujimura et al., 1975; Klingaman, 1976; Lovasik, 1983; Parisi & Falsini, 1998; Parisi, 2001; Versek et al., 2021).

Prior to discussion of ERG studies of dark adaptation over recent decades, electroretinography will be introduced more generally, together with the retinal origin of the various ERG waveform components.

## Electroretinography: neuronal basis of standard tests

**Broad neuronal types involved in retinal signal transmission.** The photoreceptor circulating current flowing in the dark depolarises the cell, leading to a steady release of glutamate at the synaptic terminal. The photoreceptor to bipolar cell synapse (reviewed recently by Burger et al., 2021) is a ribbon synapse, characterised ultrastructurally by the presence of dense regions of aligned synaptic

vesicles. Rod and cone photoreceptors hyperpolarise in response to light, reducing glutamate release at their synapse with bipolar cells. Consequent to the reduction in glutamate transmission, ON bipolar cells depolarise whilst OFF bipolar cells hyperpolarise: the sign inversion at the photoreceptor to ON bipolar cells synapse is afforded by metabotropic glutamate receptors; in contrast, OFF bipolar cells have ionotropic receptors, and respond in the same electrical 'direction' as the photoreceptors. Bipolar cells synapse with retinal ganglion cells, whose axons form the optic nerve and mostly synapse in the lateral geniculate nucleus of the thalamus. Other neurons play important roles in retinal signal processing including horizontal cells, modulating transmission at the photoreceptor to bipolar cell synapse, and the many types of amacrine cell, which can transmit signals laterally between bipolar cells as well as affecting their output to ganglion cells. There exist numerous subtypes of each of these neurons, reviewed recently by Grünert and Martin (2020).

**Representation of neuronal responses in the ERG.** The ERG represents the potential difference recorded between a recording electrode situated anteriorly (in contact with

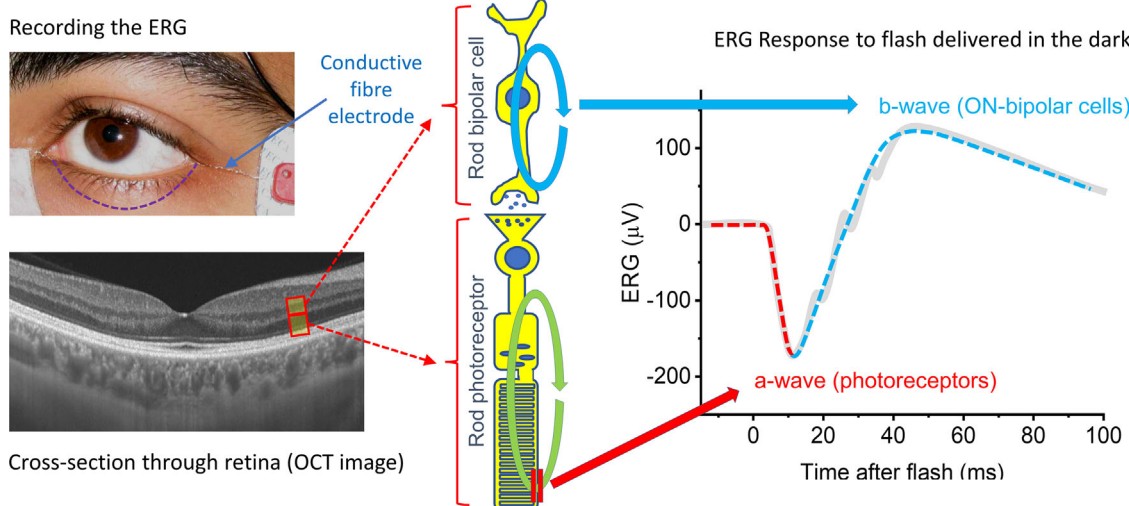

**Figure 3. ERG recording and origin of flash response components**
ERG recording is performed with a recording electrode near the anterior surface of the eye. In the upper left panel, a conductive fibre electrode is seen laterally; the electrode sits in the lower conjunctival fornix, behind the lower eyelid (denoted by the dashed purple line). The potential difference is recorded between this electrode and a skin electrode, visible on the subject's temple. A ground skin electrode is usually located on the forehead. The lower left panel shows a spectral domain optical coherence tomography (OCT) image through the central retina, showing the various layers. The red boxes highlight the position of photoreceptors and bipolar cells, shown schematically in the central part of the figure. The circulating current in the rod photoreceptor (green circular arrow) is shut off by phototransduction in response to light (outer segment membrane channels close denoted by red vertical lines), causing the photoreceptor to hyperpolarise and giving rise to the downward deflection (a-wave) of the ERG, which is shown schematically as the grey trace in the right-hand panel. Consequent reduction in glutamate release at the synapse leads to depolarising currents in the ON bipolar cell (blue circular arrow), and these give rise to the upward deflection (b-wave) of the ERG. Other cell types also contribute, including cone photoreceptors, OFF bipolar cells and amacrine cells (which are thought to give rise to the high frequency wavelets, termed oscillatory potentials seen on the ascending limb of the b-wave). The full-field ERG contains the response of the retina as a whole (not just the central part depicted in the lower left OCT image).

the cornea or the conjunctiva, or even a skin electrode placed below the lower eyelid) and a reference electrode placed posteriorly (usually a skin electrode over the temple). Broadly speaking, currents involved in neuronal hyperpolarisation and depolarisation result in negative and positive deflection, respectively. In response to a flash of light (e.g. of a strength similar to a camera flash) a negative deflection is observed lasting several milliseconds or more, followed by a positive deflection. The negative deflection is termed the a-wave, and the positive deflection is termed the b-wave. The a-wave is associated largely with the hyperpolarisation of the photoreceptors (although there is contribution from OFF bipolar cells, discussed later); the b-wave relates largely to depolarisation of the ON bipolar cells (but is also shaped by recovery of the responses from OFF bipolar cells). Figure 3 depicts ERG recording and the origin of the a-wave and b-wave components in a simplified manner.

The form of the response depends on numerous factors related to the stimulus (including spectral composition, intensity, temporal frequency, size), the subject (pupil diameter, adaptational state, presence of retinal disease) and the recording method (the types of recording electrode and their position affect the amplitude of the response). For clinical testing, the International Society for Clinical Electrophysiology of Vision (ISCEV) specifies a minimum stimulus protocol for full-field ERG testing (McCulloch et al., 2015; Robson et al., 2018), allowing evaluation of pan-retinal responses in a manner that is comparable between laboratories. Responses are recorded following pharmacological pupil dilatation. Specific stimuli are delivered following 20 min of dark adaptation (to evaluate rod-driven responses) and following 10 min of light adaptation to a standard background (to evaluate cone-driven responses). Figure 4 illustrates schematically the form of responses recorded from a healthy individual to standard clinical stimuli (standard stimulus strengths are given in the legend).

For standard clinical testing, retinal responses are thus interrogated at one of two steady states, namely (1) dark-adapted, or (2) light-adapted to a specific background luminance (namely a white background of 30 cd m$^{-2}$). However, in order to probe kinetics of retinal dark adaptation, an approach is needed whereby stimuli are delivered at multiple time intervals following extinction of a light adapting background or bleaching exposure. A bleaching exposure is one which is sufficiently strong to photo-isomerise ('bleach') a significant proportion of photopigment. Such methods were employed by the older studies mentioned in the preceding section (those conducted over the decades since the study of Karpe and Tansley, 1948). The next section will discuss findings from studies over the last 20–25 years, with reference to the experimental protocols employed and key findings.

## ERG studies of human cone and rod recovery in the dark following bright light exposures

Table 1 lists relevant studies since 1999 in chronological order, together with the broad stimulus protocols employed and findings of relevance to the current review. The majority of studies investigated recovery of specific ERG components, elicited by selected stimuli, in some cases to probe kinetics of recovery in healthy individuals,

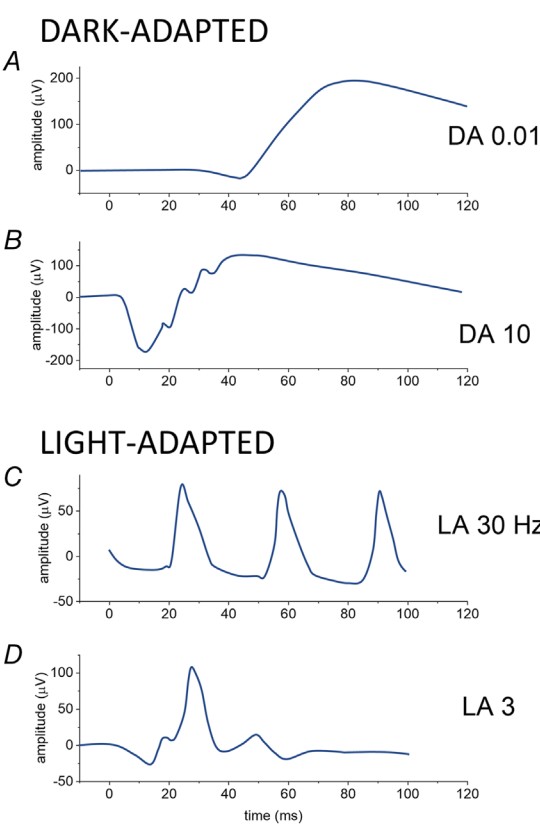

**Figure 4. Examples of ERGs (schematic traces) elicited by international standard stimuli typical of responses from a healthy individual (amplitudes correspond to those expected using a conductive fibre electrode)**
*A*, dark-adapted response to standard 'DA 0.01' dim flash (0.01 cd m$^{-2}$ s), showing a clear b-wave, but minimally detectable a-wave. The response is rod-driven. *B*, dark-adapted response to standard strong 'DA 10' flash (10 cd m$^{-2}$ s), showing a clear a-wave, b-wave and oscillatory potentials. This response is largely rod-driven, but also contains cone-driven components. *C*, light-adapted response to 30 Hz flickering stimulus (3 cd m$^{-2}$ s). *D*, light-adapted response to standard 'LA 3' photopic flash (3 cd m$^{-2}$ s), showing a-wave and b-wave. Light-adapted responses are cone-driven (rods are saturated by the light-adapting background). Dark-adapted responses (*A* and *B*) are recorded in the dark following 20 min of dark adaptation. Light-adapted responses (*C* and *D*) are recorded in the presence of a standard light-adapting background (30 cd m$^{-2}$) following 10 min adaptation to the background. An additional ISCEV standard dark-adapted ERG (elicited by a 3 cd m$^{-2}$ s flash delivered in the dark) is not shown. All stimuli and backgrounds are white, and stimulus strengths are in photopic units as is conventional for ISCEV stimuli.

**Table 1. Studies of recovery of ERG components following bleaching exposures published since 1999, with estimated bleach levels, stimulus protocols and relevant findings**

| Authors (year of publication) Rod/cone system recovery ERG component Participants Estimated bleach levels | Stimulus protocol | Relevant findings |
|---|---|---|
| Thomas & Lamb (1999) Rod recovery Rod-driven a-wave Healthy participants Estimated bleach levels: 8–100% | Dim and bright blue flashes. Removed cone contribution by subtraction of responses to photopically matched red flashes. Response to bright flash allowed estimation of rod circulating outer segment current. Response to dim flash allowed estimation of 'amplification constant' of phototransduction. | Following a full bleach, rod photoreceptor circulating current, as estimated by the bright flash a-wave, took 30 min to recover to dark-adapted levels, with no detectable response in the first 5 min, and half-recovery complete in 13–17 min. Time taken to recover to fixed proportion of final level increased linearly with fractional bleach over a range of bleaches, resembling aspects of psychophysical recoveries. Recovery of the dim-flash response showed no evidence of a detectable change in amplification of phototransduction after adjustment for reduction in quantal catch due to bleaching. |
| Paupoo et al. (2000) Cone recovery Cone-driven a-wave Healthy participants Estimated bleach levels: ∼50% and ∼99% | Dim and bright red flashes pre- and post-bleach. Rods largely saturated by presence of blue rod-saturating background. | Cone photoreceptor circulating current, as estimated by the bright flash a-wave, recovered to dark-adapted levels within 30 s following a near-total bleach, and within 3 s following an ∼50% bleach. Dim-flash a-wave response, measured at a fixed time after the flash, was assumed to be proportional to pigment levels. Recovery could be fitted with an exponential of time constant of ∼1.5 min following a near total bleach, or ∼0.7 min following an ∼50% bleach. Limitations included inability to assess circulating current at very early time points following bleaching exposure and assumptions of linearity in dim-flash response-intensity in deriving pigment levels. |
| Mahroo & Lamb (2004) Cone recovery Cone-driven a-wave Healthy participants Estimated bleach levels: 27–96% | Dim red flashes before and after bleaching exposures. Rods saturated by presence of blue rod-saturating background. | Flash response amplitudes post-bleach were transformed according to the response–intensity relation to estimate pigment levels. Pigment recovery following a range of bleaches appeared to more closely fit a 'rate-limited' expression whereby recovery proceeds at a common initial linear rate, rather than as an exponential recovery. The novel 'rate-limited' expression was also found to provide a better fit to previous studies of cone pigment regeneration using densitometry (although with a different value for the rate parameter). The authors suggested the rate limit could ensue from a physical process whereby chromophore diffuses from a fixed pool (Lamb & Pugh, 2004; Mahroo & Lamb, 2004). A later analysis showed that such kinetics could also ensue from an enzymatic rate limit (Lamb et al., 2015). |

*(Continued)*

**Table 1. (Continued)**

| Authors (year of publication) Rod/cone system recovery ERG component Participants Estimated bleach levels | Stimulus protocol | Relevant findings |
|---|---|---|
| Kenkre et al. (2005) Cone recovery Cone-driven a-wave Healthy participants Estimated bleach levels: ∼90% | Bright white flashes delivered at sub-second intervals following extinction of steady-state bleaching exposure. Flash strength adjusted to take into account reduction in quantal catch. Dark-adapted cone responses estimated by recording on dim blue rod-saturating background. | Cone photoreceptor circulating current, as estimated by the bright flash a-wave, recovered substantially within 20 ms and fully within 100 ms following extinction of a steady-state bleaching exposure (estimated to bleach 90% of cone photopigment). This is approximately 50,000 times faster than recovery in rods following a similar bleach, and suggests that cone photoreceptor circulating current, unlike rod current, appears not to be suppressed by bleaching products. |
| Binns & Margrain (2005) Cone recovery Focal 41 Hz flicker ERG, 1st harmonic Healthy participants Estimated bleach levels: 86% | Amber 41 Hz flickering stimulus, subtending 20 degrees, within luminance-matched surround; ERGs recorded at 20 s intervals for 4 min following 2 min bleaching exposure. | ERG first harmonic amplitudes recovered within 4 min to pre-bleach levels. Recoveries could be fit with single exponentials (time constants ranged from 0.51 to 2.13 min) or rate-limited expressions. Recoveries expected to reflect time course of cone pigment regeneration, but amplitudes are not necessarily linearly related to pigment levels. |
| Cameron et al. (2006) Rod recovery Rod-driven b-wave Healthy participants Estimated bleach levels: 1.5–96% | Dim blue flashes delivered pre- and post-bleach to evaluate rod-driven responses. Very dim background used to minimise negative scotopic threshold response. Responses to the same flashes on different backgrounds were used to derive an estimate of an 'equivalent background' post-bleach. | The decay of the 'equivalent background' post-bleach showed similar kinetics to the S2 and S3 components of psychophysical dark adaptation. The authors suggested the S2 component may arise from unregenerated opsin and the S3 component could arise from closure of ion channels by all-*trans*-retinal. |
| Binns & Margrain (2007) Cone recovery Focal 41 Hz flicker ERG, 1st harmonic Patients with age-related maculopathy and age-matched controls Estimated bleach levels: 86% | Amber 41 Hz flickering stimulus, subtending 20 degrees, within luminance-matched surround; ERGs recorded at 20 s intervals for 5 min following 2 min bleaching exposure. | Recovery of amplitude post-bleach was fitted with single exponentials. Rate of recovery was defined as the reciprocal of the time constant. The rate was on average significantly slower in patients compared with controls. |
| Cameron et al. (2008) Rod recovery Rod-driven b-wave Healthy participants Estimated bleach levels: ∼2–99% | Dim blue flashes delivered pre- and post-bleach to evaluate rod-driven responses. Flashes delivered post-bleach to achieve a criterion response amplitude. Comparison with responses on different backgrounds to derive estimates of equivalent backgrounds. | Analysis of time to peak of b-wave responses to flashes chosen to elicit a criterion response amplitude post-bleach. Findings suggested post-bleach sensitivity is consistent with 'equivalent background' and an additional 'pure desensitisation' component. The equivalent background component declined with the S2 slope seen in psychophysical recovery, and was thought to reflect the decline in unregenerated opsin. |
| Ruseckaite et al. (2011) Rod recovery Rod-driven b-wave Healthy participants Estimated bleach levels: ∼2–95% | Dim blue flashes delivered pre- and post-bleach to evaluate rod-driven responses and to determine decay of equivalent background, compared with psychophysical recoveries in the same participants. | ERG recoveries as described in previous study (Cameron et al., 2008). Comparison with psychophysical recoveries showed similarities: lateral shifts in recoveries appeared to reflect the same time course of decline in unregenerated opsin at high bleach levels (high levels of unregenerated opsin), but showed discrepancy at lower levels of unregenerated opsin. |

*(Continued)*

**Table 1. (Continued)**

| Authors (year of publication)<br>Rod/cone system recovery<br>ERG component<br>Participants<br>Estimated bleach levels | Stimulus protocol | Relevant findings |
|---|---|---|
| Wood et al. (2011)<br>Cone recovery<br>Focal 41 Hz flicker ERG, 1st harmonic<br>Healthy participants<br>Estimated bleach levels: 86% | Amber 41 Hz flickering stimulus, subtending 20 degrees, within luminance-matched surround; ERGs recorded at 20 s intervals for 5 min following 2 min bleaching exposure ('equilibrium' bleach) or single bright flash ('photoflash' bleach). | Recovery of amplitude was fitted with single exponential recoveries with mean time constants of 117 and 112 s for equilibrium and photoflash bleach recoveries respectively. Recoveries following equilibrium bleach showed more repeatability than following a flash bleach. Rate of recovery (from equilibrium bleach) showed significant correlation with age, with slower recovery in older participants. |
| Freund et al. (2011)<br>Rod and cone recovery<br>Rod and cone-driven a-waves and b-waves<br>Healthy participants<br>Bleach level not estimated | Dim and bright white flashes and a red flash delivered at regular intervals in the dark following exposure to multiple strong flashes on international standard (30 photopic cd m$^{-2}$) light-adapting background. | Dim-flash ERG b-wave amplitudes steadily increased during dark adaptation. Peak times also increased, reaching a plateau around 8 min. For brighter white flash, a-wave and b-wave amplitudes increased during dark adaptation; a-wave peak times decreased reaching a plateau within the first 5 min, whilst b-wave peak times increased, reaching a plateau at 10–15 min. For red flashes, a-wave amplitudes decreased slightly initially and then remained steady during dark adaptation, whilst b-wave amplitudes increased more steadily; a-wave peak times increased (less evident in older participants) and b-wave peak times also increased.<br>Generally, peak times were earlier and amplitudes greater in younger, compared with older, age groups. |
| Mahroo & Lamb (2012)<br>Cone recovery<br>Cone-driven a-wave<br>Healthy participants<br>Estimated bleach levels: ∼99% | Delivered dim red flashes before and after very intense bleaching exposures. Rods saturated by presence of blue rod-saturating background. | Response amplitudes post-bleach transformed to generate estimates of photopigment regeneration as in previous study (Mahroo & Lamb, 2004). Recovery showed initially linear rate, with rate-limited expression providing a markedly superior fit compared with exponential recovery. However, recovery rate was slower than for previous investigation of smaller bleaches, suggesting slowing of recovery rate following very intense exposures. Recovery kinetics were very similar to previous densitometric measurements of pigment, consistent with the ERG-derived recovery reflecting photopigment levels. |
| Kubota et al. (2012)<br>Rod recovery<br>Rod-driven b-wave<br>Healthy participants taking emixustat<br>Bleach level not estimated | Dim white flashes delivered before and after 10 min exposure to full-field bleaching light (556 cd m$^{-2}$). | In subjects receiving placebo, response amplitudes recovered to within 90% of prebleach levels within 20 min post-bleach. Participants receiving the visual cycle modulator showed slowing of recovery with increasing dose of drug. Maximal effects were seen on day 2, with recovery to baseline on day 7 after drug administration. |

*(Continued)*

**Table 1. (Continued)**

| Authors (year of publication)<br>Rod/cone system recovery<br>ERG component<br>Participants<br>Estimated bleach levels | Stimulus protocol | Relevant findings |
| --- | --- | --- |
| Dimopoulos et al. (2013)<br>Rod recovery<br>Rod-driven b-wave<br>Healthy participants and patients with AMD<br>Bleach level not estimated | Dim white flashes delivered at regular intervals (2 min) in the dark following 10 min exposure to international standard (30 photopic cd m$^{-2}$) light-adapting background as well as multiple strong flashes delivered on this background. | Dim-flash b-wave amplitudes recovered with time in the dark. Recovery was slower on average (as quantified by time to half recovery) in eyes with dry or wet AMD compared with control eyes. |
| Dugel et al. (2015)<br>Rod recovery<br>Rod-driven b-wave<br>AMD patients taking emixustat<br>Bleach level not estimated | Dim white flashes delivered before and after 3 min exposure to full-field bleaching light (∼500 cd m$^{-2}$). | Patients showed slowing of post-bleach recovery of rod-driven ERG amplitudes: the slowing of rate of recovery ranged from 34% to 90% in the 2 mg and 10 mg (lowest and highest dose) treatment groups respectively, as measured at day 14 of treatment. |
| Alim-Marvasti et al. (2016)<br>Rod recovery<br>Rod-driven b-wave<br>Healthy participant<br>Bleach level not estimated | Dim white flashes delivered at regular intervals in dark following monocular viewing of smartphone screen. | Response amplitudes attenuated in eye that had viewed the smartphone, and then recovered following several minutes in the dark to match those of the fellow eye. Demonstration that 'blindness' experienced by patients following monocular smartphone viewing was related to physiological retinal adaptation, demonstrable at the level of rod-driven bipolar cell responses. |
| Kubota et al. (2022)<br>Rod recovery<br>Rod-driven b-wave<br>Stargardt patients taking emixustat<br>Bleach level not estimated | Dim white flashes delivered before and after 3 min exposure to bleaching light. | As in previous studies of emixustat, patients showed slowing of rate of post-bleach recovery of rod-driven ERG amplitudes, showing a dose response, consistent with an effect on slowing of retinoid cycle. |

or to develop tests for possible clinical use, or for assessing effects of drug treatment. Freund et al. (2011) measured responses to multiple ERG stimuli in the dark following exposure to strong flashes on the standard light-adapting background. There have also been clinically important studies (Bach et al., 2020; Hamilton & Graham, 2016), not included in the table, comparing the effects of shorter and longer periods of dark adaptation on the amplitude and peak time of standard stimuli, with a view to investigating the effect of shorter adaptation times on standard clinical testing.

A very early component (latency of a fraction of a millisecond) that can be present in the ERG elicited by bright flashes deserves mention. This was first identified in intra-retinal recordings (Brown & Murakami, 1964), and is termed the early receptor potential (ERP). It arises from charge movements from conformational changes of opsins in the plasma membrane in response to photo-isomerisation of the chromophore. Rhodopsin within the disc membranes does not contribute to this potential.

Goldstein and Berson investigated recovery of the human ERP during dark adaptation in healthy individuals and in participants with hereditary retinal disease (Berson & Goldstein, 1970; Goldstein & Berson, 1969). However, recording this component can be challenging, and we are not aware of recent studies of ERP recovery following bleaches in humans, so the ERP will not be discussed further.

In the following sections, studies will be grouped according to ERG component investigated (a-wave, b-wave, flicker ERG), not necessarily in chronological order. Insights into pigment regeneration kinetics will be discussed, as well as relation to aspects of psychophysical recovery, where relevant.

**Rod-driven a-wave amplitudes demonstrate prolonged suppression of circulating current post-bleach.** Thomas & Lamb (1999) delivered dim and bright flashes in the dark pre- and post-bleach (and in the presence of

different backgrounds) to evaluate human rod responses. A common challenge in investigations of a particular class of photoreceptors (rods or cones) is intrusion of responses from the other class. In this study, they chose blue flashes (to favour stimulation of rods over L- and M-cones) and also delivered photopically matched red flashes to elicit the L- and M-cone-driven contribution to the blue-flash responses. By subtracting the red-flash responses from the blue-flash responses, they were able to derive an estimated isolated rod-driven response. They investigated the following parameters of rod photo-transduction: the rod circulating current referred to the ionic current entering the photoreceptor outer segments (which is shut off by light); the 'amplification constant' is a parameter effectively quantifying the gain in the photo-transduction cascade. They employed two approaches to derive estimates of these parameters from a-wave measurements, namely the fitting of a mathematical model based on kinetics of the activation steps of photo-transduction (Lamb & Pugh, 1992) and the measurement of responses at fixed post-flash times.

Following a full bleach, they found that rod circulating current (as estimated from the a-wave elicited by bright flashes) was undetectable in the first 5 min and took 30 min to recover to dark-adapted levels. By interrogating responses to dim and bright flashes, they found no evidence of a change in the amplification constant after adjustment of changes in quantal catch (this term refers to the available photopigment for photoisomerisation by light) post-bleach. They also used responses obtained on different backgrounds pre-bleach to derive estimates of an 'equivalent background' that would give similar response amplitudes to those they recorded post-bleach. This procedure has been termed a 'Crawford transformation' in the psychophysical literature (Blakemore & Rushton, 1965; Barlow, 1972; Crawford, 1947). They found that the decay in equivalent background intensity post-bleach bore some similarity to psychophysical recoveries (such as those shown in Fig. 1): parallel slopes were observed over a portion of the recovery, and the time taken to reach a particular threshold recovery showed a linear relationship with bleaching level. They suggested that the presence of photoproducts post-bleach that underwent 'rate-limited' removal (as suggested by Lamb, 1981) might underlie both (psychophysical and electrophysiological) phenomena. This was later proposed to be unregenerated opsin (Lamb & Pugh, 2004), i.e. opsin prior to re-binding with 11-*cis*-retinal.

**Cone-driven a-wave amplitudes indicate rapid recovery of circulating current, with slower, rate-limited recovery of photopigment.** Selective investigation of cone-driven ERG responses requires removal or suppression of rod-driven components. Paupoo et al. (2000) delivered red flashes (to favour L- and M-cone stimulation over

rods) in the presence of a rod-saturating background. The background was blue to favour saturation of rods over L- and M-cones, with an intensity chosen ($\sim$2–3 photopic and $\sim$30 scotopic cd m$^{-2}$ through a pharmacologically dilated pupil; higher strength when natural pupils were used) to minimally light-adapt the cones. The standard white ISCEV photopic background, for comparison, is 30 photopic cd m$^{-2}$.

The authors delivered flashes on the rod-saturating background before and after intense bleaching exposures. These were bright and dim flashes (measuring the a-wave at fixed post-flash times in both cases) to probe recovery of cone circulating current and cone photopigment, respectively. Based on the electrical responses of single photoreceptors, the amplitude of the response to a bright flash (that transiently but fully shuts off the photoreceptor circulating current) will give an estimate of the circulating current. The basis of the use of the dim-flash a-wave to estimate photopigment will be discussed subsequently. The bright-flash a-wave amplitude recovered within 30 s following an almost total bleach, and within a few seconds following a partial bleach, indicating a recovery of cone circulating current that is orders of magnitude faster than in rods following similar bleaching exposures.

Kenkre et al. (2005) investigated recovery of cone circulating current at very early times following steady state exposures estimated to bleach $\sim$90% of cone pigment. They used flashes calculated to produce similar numbers of photoisomerisations as those delivered pre-bleach (i.e. adjusting for the change in quantal catch), so that meaningful comparison could be made. One potential challenge with the use of the ERG a-wave to derive estimates of cone photoreceptor responses is the significant contribution to the cone-driven a-wave that has been shown to originate in cone-driven OFF bipolar cells: this has been shown in macaque (Robson et al., 2003), and is likely to apply to human ERGs also. These post-receptoral signals comprise much of the dim-flash cone-driven a-wave, and for bright flashes, such contributions can begin within 5 ms post-flash. Kenkre et al. thus measured a-waves at a very early time post-flash to avoid significant post-receptoral intrusion. They estimated that cone circulating current recovered fully within 100 ms following extinction of a steady state exposure that gave a 90% bleach, and substantial recovery was seen within 20 ms. Such recovery is some 50,000 times faster than recovery of current in rods following similar bleaches, and suggests that products of bleaching (unregenerated opsin), present following extinction of the bleaching exposure, do not substantially shut off circulating current in cones, in contrast to the situation in rods.

In the study of Paupoo et al. (2000), the dim-flash cone-driven a-wave recovered over several minutes, consistent with this reflecting the kinetics of cone

photopigment regeneration. Based on the analysis of Lamb & Pugh (1992), the response to a dim flash, measured at early post-flash times, is expected to be proportional to the product of three factors, namely the effective flash strength (which, for a constant-strength flash, will in fact be proportional to photopigment levels), the amplification constant of (the activation phase of) phototransduction, and the circulating current (Lamb & Pugh, 1992). If cone circulating current recovers very quickly (as shown from bright-flash responses), and if the amplification constant is assumed to be unchanged (as appeared to be the case for rods from the investigations of Thomas & Lamb, 1999), then the response amplitude would reflect photopigment levels; this was the basis of the conclusions of Paupoo et al. (2000) in relation to photopigment recovery.

However, the flash strengths used by Paupoo et al. (2000), though dim, are outside the linear range, such that response amplitude as measured is not proportional to flash strength, and therefore will not be proportional to pigment levels. Direct modelling of photopigment kinetics requires accounting for the non-linearity in the response–intensity relation. Specific adjustment for this non-linearity was performed by Mahroo & Lamb (2004). This is shown schematically in Fig. 5 with the procedure explained in the legend. This allowed post-bleach dim-flash response amplitudes to be transformed into estimates of unbleached photopigment. The similarity between the form of post-bleach responses and pre-bleach responses to weaker flashes supported the validity of this approach. The substantial post-receptoral contribution to the dim-flash a-wave was also considered: the authors used a similar approach using much brighter flashes (measuring at very early post-flash times before significant post-receptoral contributions), and found that the estimated recovery of cone pigment was similar to that found using the dim-flash approach.

Mahroo & Lamb (2004) found that estimated cone pigment regeneration appeared to proceed at an initial constant linear rate after a wide range of initial bleaches: such behaviour was not consistent with a first order process (where recovery rate was proportional to pigment bleached), but with a 'rate-limited' process, similar to that postulated by Lamb (1981) to underlie a component of the psychophysical recovery of rod-mediated sensitivity. Figure 6 (upper and middle panel) illustrates the difference between the two forms of recovery. In a first order process, where recoveries proceed as single exponentials with a fixed time constant, the initial rate of recovery is faster following larger bleaches. With rate-limited recovery, the initial rate is constant for a range of bleaches, such that recoveries following these bleaches proceed as parallel lines. As well as the initial bleach level, recovery is governed by two parameters, the initial rate following a total bleach, $v$, and

the bleach level, $K_m$, at which recovery is half the maximal rate. Mahroo & Lamb (2004) also found that bleach levels as a function of exposure duration (for non-steady state exposures) and as a function of bleaching intensity (for steady-state exposures) were more consistent with rate-limited, than with first order, recovery kinetics.

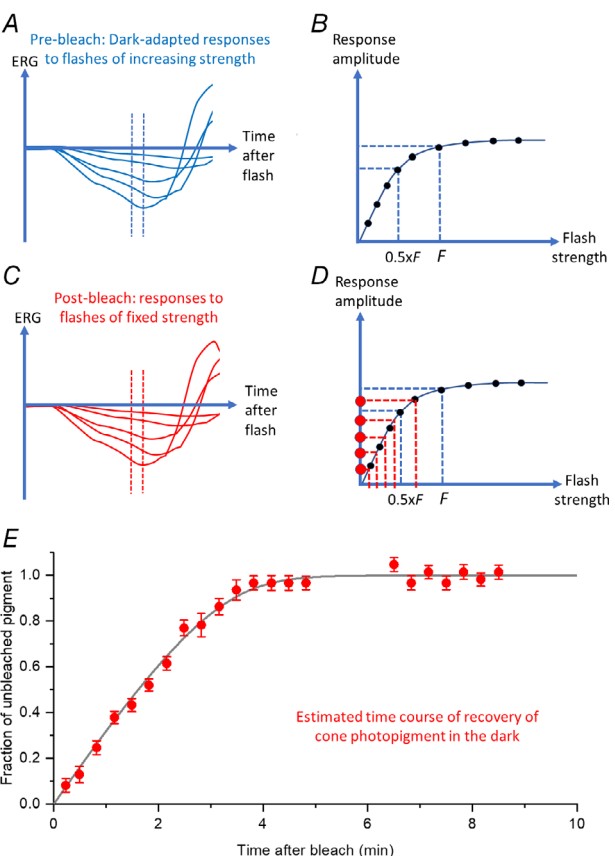

**Figure 5. Method of transformation of post-bleach cone-driven a-wave amplitudes to derive estimates of proportion of unbleached photopigment**

*A*, schematic representation of ERG a-wave responses to flashes of various strengths delivered in the presence of a dim blue rod-saturating background. Larger amplitude responses are elicited by stronger flashes. *B*, response amplitudes from *A* (averaged over a fixed post-flash time window denoted by vertical dashed lines in *A*) plotted against flash strength, to yield a response–intensity relation. This can be seen to be non-linear. The response elicited by a flash of strength 0.5*F* (i.e. a flash of half the strength of a given flash strength *F*) is more than half the response elicited by flash strength *F*. *C*, schematic representation of responses to flashes of a fixed strength delivered at progressive times post-bleach. *D*, response amplitudes in *C* are used to derive an 'effective' flash strength by inversion of the response–intensity relation. Dividing the effective flash strength by the actual flash strength yields an estimate of the fraction of unbleached photopigment. The assumptions involved, and the likelihood of their validity, are detailed by Mahroo & Lamb (2004). *E*, fraction of unbleached pigment (as estimated by this technique) plotted against post-bleach time following an intense bleaching exposure. The initial recovery appears to be linear with time, consistent with 'rate-limited' kinetics. These data are replotted from Mahroo & Lamb (2012).

In addition, rate-limited kinetics appeared to provide a closer fit to previous measurements of cone pigment regeneration made using reflection densitometry.

It was proposed that such kinetics might emerge from diffusion of 11-*cis*-retinal (into photoreceptor outer segments) through a resistive barrier from a larger pool, possibly within the retinal pigment epithelium (Lamb & Pugh, 2004; Mahroo & Lamb, 2004). The retinoid cycle was analysed in detail by Lamb & Pugh

(2004), and the rate-limited model was termed the MLP (Mahroo & Lamb, 2004; Lamb & Pugh, 2004) model of pigment regeneration. Delivery of 11-*cis*-retinal to cones restored sensitivity by making more pigment available to catch photons. Delivery of 11-*cis*-retinal to rods restored sensitivity by removing free opsin (i.e. combining with it to form rhodopsin), and thus terminating the action of opsin in shutting off rod photoreceptor circulating current. This rate-limited removal of opsin could also explain the behaviour of the S2 component of psychophysical recovery. Lamb & Pugh (2004) demonstrated that the model was consistent with both psychophysical and densitometry data in healthy individuals and in a range of clinical conditions with abnormal dark adaptation.

Mahroo & Lamb (2012) later showed that cone pigment regeneration following very intense bleaching exposures, as estimated from recovery of the cone-driven dim-flash a-wave, proceeded at a slower rate than they had found previously for less intense exposures (Mahroo & Lamb, 2004). The initial rate of recovery was still clearly linear, with the rate-limited expression providing a much closer fit than an exponential recovery. They speculated the slower rate might reflect depletion of the pool of retinoid, from which diffusion was occurring. They also showed a very clear similarity between ERG-derived and (previously published) densitometric estimates of cone pigment regeneration following similar intensity exposures. The densitometric measurements were from the study of van Norren & van de Kraats (1989). These are replotted in Fig. 7, together with the rate-limited model curve fitted to the ERG data: the curve also fits the densitometry data, and the two methods provide similar estimates of pigment.

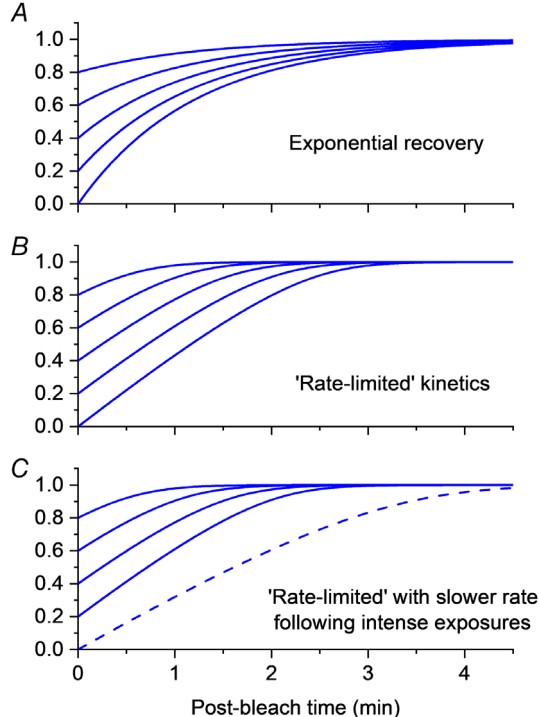

**Figure 6. First order and 'rate-limited' kinetics of photopigment regeneration**

*A*, simulated recovery of pigment following a range of bleach levels (*y*-axis denotes fraction of unbleached pigment) assuming first order kinetics. All recoveries proceed as exponentials with a common time constant. *B*, simulated recoveries in the case of an initial 'rate-limit' whereby recovery proceeds at a common initial linear rate following a range of bleaches. These recoveries are governed by two parameters, *v* and $K_m$, explained in the text. Such kinetics appear to be consistent with a range of estimates of rod and cone pigment regeneration, measured using different approaches, and can explain features of psychophysical recovery of scotopic vision. *C*, rate-limited kinetics, but with a slowing of rate following very intense bleaches. This was found for cone pigment regeneration by Mahroo and Lamb (2012) as estimated using the ERG. Interestingly, the slower rate parameter also provided a good fit to multiple densitometric studies of cone pigment regeneration following similarly intense bleaching exposures. The origin of the rate-limit was suggested to be a resistive barrier to diffusion of retinoid (Lamb & Pugh, 2004; Mahroo & Lamb, 2004); later, an enzymatic rate limit was proposed (Lamb et al., 2015). The time axis relates to cone pigment recovery; rod pigment recovery is more slow, but follows a similar rate-limited expression, with a similar $K_m$ parameter (ranging from 0.15 to 0.2), but a slower value for *v* (approximately 0.085 min$^{-1}$ in rods, compared with 0.3–0.5 min$^{-1}$ in cones).

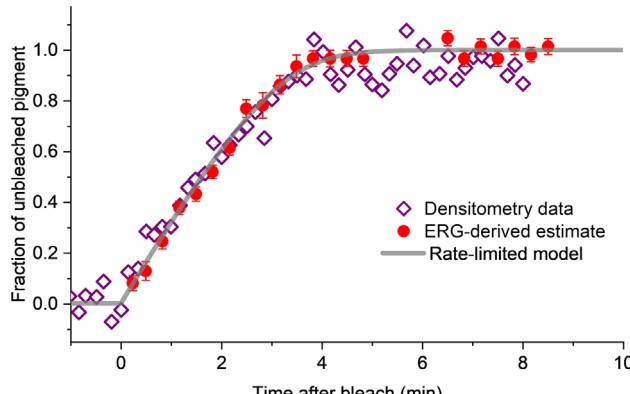

**Figure 7. Comparison of densitometry and a-wave-derived estimates of cone pigment regeneration following intense bleaching exposures**

The densitometry data are replotted from van Norren & van de Kraats (1989), normalised to the final level, and the ERG-derived estimates are replotted from Mahroo & Lamb (2012), with the MLP rate-limited model also shown (grey curve): the model appears to provide a reasonable fit to both sets of data points.

The possible basis of rate-limited kinetics was re-evaluated by Lamb et al. (2015), who showed that such kinetics also emerged under conditions of an enzymatic rate limit (rather than a resistive limit to diffusion). Although both mechanisms yielded similar kinetics, they presented theoretical arguments favouring an enzymatic limit. Overall, a broad range of ERG-derived, psychophysical and densitometric data appear to be consistent with rod and cone pigment regeneration proceeding according to rate-limited kinetics: a $K_m$ parameter of $\sim$0.15–0.20 appears to fit both rod and cone regeneration; a rate, $v$, parameter of approximately 0.3–0.5 min$^{-1}$ appears to fit cone recoveries, whilst a rate of approximately 0.08–0.095 min$^{-1}$ seems to fit rod recoveries (Lamb & Pugh, 2004, 2006; Lamb et al., 2015; Mahroo & Lamb, 2004; Mahroo & Lamb, 2012).

**Rod-driven b-wave amplitudes show desensitisation consistent with effects of an equivalent background and have been used to probe recovery in physiological and pathological states.** Dim flashes delivered in the dark elicit a b-wave, representing the depolarisation of rod-driven ON-bipolar cells (as shown in Fig. 4*A*). Although this is in response to light-induced hyperpolarisation of the rods (and consequent reduction in glutamate release at the photoreceptor-to-bipolar cell synapse), the a-wave is not usually visible with very dim flashes, partly due to the photoreceptor hyperpolarisation being hidden by the larger b-wave. Cameron et al. (2006) recorded responses to dim flashes (blue to favour stimulation of rods over L- and M-cones) delivered in the dark and on backgrounds of different intensities, and also before and after a range of bleaching exposures. They found that responses post-bleach were of smaller amplitude, and shortened peak time, compared with those obtained pre-bleach. With time, post-bleach responses recovered to the pre-bleach shape (increasing in amplitude and peak time). The post-bleach responses showed similarity to those obtained pre-bleach, but on backgrounds of different intensities. They were able to apply a 'Crawford transformation', as explained above, to generate estimates of an 'equivalent background'. The intensity of this background declined with time post-bleach showing behaviour similar to the S2 and S3 components of psychophysical recovery.

This behaviour was examined in more detail in a subsequent study (Cameron et al., 2008), where flashes were selected to elicit a criterion response amplitude at progressive post-bleach times, and a mathematical model was applied to the responses to allow more reliable identification of the response kinetics (time to peak) in the midst of noise. They found post-bleach desensitisation appeared to show two components, an 'equivalent background' component that declined with a time course close to the S2 component of psychophysical recovery, and a 'pure desensitisation' component declining with a time course similar to the S3 component. The S2 component was more clearly identified, and appeared to correlate with the recovery of kinetics of the response. The authors also suggested the faster kinetics of earlier post-bleach responses might relate to the faster temporal resolution of rod-driven vision in the presence of backgrounds compared with the fully dark-adapted state.

Comparison of rod-driven b-wave recoveries and psychophysical recovery following bleaches was performed in the same subjects by Ruseckaite et al. (2011). Estimates of equivalent background were generated by both approaches, and interestingly although an 'S2' component was identified by both modalities, the slopes of the recoveries differed, and the equivalent background estimated by the b-wave was a log unit higher than that derived psychophysically. Nevertheless, the two methods showed agreement at higher bleach levels, consistent with both measurement approaches reflecting the concentration of unregenerated opsin, but the discrepancy appeared when levels of opsin were low (i.e. rhodopsin levels almost fully recovered).

Subsequent studies by other authors have made use of the rod-driven b-wave response to dim flashes as a method of quantifying recovery of the rod system in the dark, but without generating estimates of equivalent backgrounds. Dimopoulos et al. (2013) delivered dim flashes (corresponding to the ISCEV standard dim flash) following 10 min exposure to the standard ISCEV light-adapting background (with strong flashes also having been delivered on this background). They fitted an empirical sigmoid function to recovery of b-wave amplitudes in the dark. They found that on average patients with age-related macular degeneration (AMD) showed slower recovery than control participants, and there was no difference found between eyes with wet or dry AMD. This is consistent with studies that have shown delays in recovery of psychophysical rod-mediated sensitivity in AMD (reviewed recently by Murray et al. (2021), and the study of Dimpopoulos et al. interestingly shows that such a delay is detectable in panretinal ERG responses. In AMD, the RPE and photoreceptors are affected, with deposits (drusen) building up beneath the RPE and other deposits (pseudodrusen) developing between the photoreceptors and the RPE, and so the retinoid cycle is likely to be affected. The numerous conditions in which drusen and drusen-like deposits occur were reviewed by Khan et al. (2016), and include visual cycle disorders.

Alteration in the recovery of the rod-driven dim-flash b-wave post-bleach has also been used as a proof of principle to show that inhibitors of the retinoid cycle have a detectable and quantifiable effect in the retina *in vivo*, and to establish the duration and dose relationship of such an effect. Emixustat inhibits the function of

RPE65 (involved in the visual cycle, Fig. 2) and is being trialled as a treatment for a number of retinal diseases, including Stargardt disease, which arises from bi-allelic pathogenic variants in the *ABCA4* gene. In this condition, toxic by-products of the retinoid cycle build up, and animal studies have suggested that slowing the visual cycle can prevent or slow the rate of retinal degeneration (Radu et al., 2003). Table 1 includes a number of studies examining the effect of emixustat on recovery of the dim-flash b-wave following bleaching exposures in healthy participants (Kubota et al., 2012), patients with AMD (Dugel et al., 2015) and patients with Stargardt disease (Kubota et al., 2022).

The dim-flash response following monocular light exposure can be used to objectively and quantifiably demonstrate reduction in sensitivity of one eye compared to the fellow dark-adapted eye. Alim-Marvasti et al. (2016) reported two cases of patients reporting recurrent visual loss in one eye. Clinical investigations did not reveal a cause, but a careful history revealed that both patients had been inadvertently viewing a smartphone monocularly whilst lying in bed; as they were lying on their sides, the other eye was covered by a pillow. Thus, the viewing eye was light-adapted, whilst the covered eye became dark-adapted. Subsequently, when looking around in a dimly lit room, they noticed that their vision appeared inexplicably poorer in one eye (the light-adapted eye). The authors termed the condition transient smartphone 'blindness', and reported these cases to inform the medical community so that potential anxiety (for both physician and patient) and extensive, sometimes invasive, investigations can be avoided once a typical history (of monocular light adaptation, and subsequent recovery of sensitivity in the light-adapted within an appropriate time frame) has been elicited. The authors demonstrated in a healthy individual, following viewing of a smartphone screen for 20 min with one eye covered, that the rod-driven dim-flash b-wave was reduced in the eye that had viewed the screen, compared with the covered (dark-adapted eye). With time in the dark (over the course of 20 min or so), the amplitude recovered in the eye that had viewed the phone, to equal that recorded from the fellow eye. The findings demonstrated that the localised illumination from a smartphone could lead to a prolonged reduction in sensitivity, demonstrable at the level of the rod bipolar cell signals in full-field ERGs, and the phenomenon is physiological, not pathological.

**Photopic flicker ERG recovery: effects of age and disease on cone system recovery following bleaches.** Recovery of the cone-driven flicker ERG following bleaches has been investigated. Binns & Margrain (2005) recorded ERG responses to a 41 Hz focal (subtending 20 degrees) amber flickering stimulus, within a steady surround of matched luminance. The flicker frequency is beyond the temporal resolution of rods, and the amber colour favours L- and M-cone stimulation. Although the response is cone-driven, the ERG is likely to arise from signals in cone-driven bipolar cells. The first harmonic of the response was analysed, and the stimulus was delivered at 20 s intervals for 4 min following a 2 min bleaching exposure. They found that recovery of amplitude post-bleach could be fit by a single exponential recovery (with time constants in healthy participants ranging from 0.51 min to over 2 min) and also by the rate-limited expression developed by Mahroo & Lamb (2004). The amplitude is not known to be linearly related to photopigment levels (or stimulus intensity), and hence the amplitude will not give a direct estimate of photo-pigment recovery kinetics. However, the time course is likely to indirectly reflect pigment regeneration, and hence the test could reflect the health of the outer retina (cones and their visual cycle) in the central 20 degrees of retina.

In a later study, the authors showed that the rate of recovery was slower on average in patients with age-related maculopathy compared with age-matched controls (Binns & Margain, 2007). Wood et al. (2011) compared recoveries measured with this method in healthy participants following either a 2 min bleaching exposure or following a single bright bleaching flash. They found that both recoveries appeared to have similar rates, but recoveries were more repeatable following the 2 min bleaching exposure. They also found significantly slower recoveries in older participants, consistent with possible slowing of the cone visual cycle with age.

## Concluding remarks and future directions

**The ERG as a tool to probe physiological processes underlying human dark adaptation.** ERG studies of recovery of human retinal sensitivity following bleaching exposures have contributed significantly to our understanding of recovery of rod- and cone-mediated vision. They have provided evidence of 'rate-limited' recovery of both rod and cone photopigment, and have shown cone photo-receptor circulating current to be 'immune' to the presence of high levels of photoproducts such that even with 90% of the pigment bleached (opsin unbound to 11-*cis*-retinal), current returns to dark-adapted levels within a fraction of a second following extinction of a steady-state bleaching background. This is in stark contrast to recovery of circulating current in rods, which occurs orders of magnitude more slowly, likely owing to the action of unbound opsin in activating the photo-transduction cascade and shutting off current.

Whilst bleached photopigment has been shown to activate transduction in both rods and cones (Cornwall et al., 1995; Matthews et al., 1996), the molecular basis for the difference in effect on circulating current following a bleach remains to be fully elucidated. Single-cell

recordings have shown that mouse M-cones can maintain circulating current near the dark-adapted level even with more than 90% of opsin in the bleached state (Nikonov et al., 2006), consistent with findings from the human ERG studies described above (Kenkre et al., 2005). Cone phototransduction has lower amplification than rods, with shorter lifetimes of active photopigment, and faster inactivation of activated transducin (Lobanova et al., 2010; Nikonov et al., 2006). The lower overall gain in the transduction cascade in cones might play a role in allowing maintenance of circulating current similar to dark levels even in the presence of high levels of opsin post-bleach.

A number of additional questions remain and invite future investigation. Although significant progress has been made in linking aspects of pigment regeneration with retinal electrophysiological and visual psychophysical recovery, the precise mechanisms underlying the elevation of psychophysical threshold and the contribution of processes occurring at different levels of the retina are still to be fully elucidated. The relative contribution of the different (retinal Müller cell and RPE) pathways to cone pigment regeneration in humans is not known. Proteins involved in phototransduction (including transducin and arrestin) are translocated between outer and inner segments during light and dark adaptation (Sokolov et al., 2002). Translocation of transducin can help rods avoid saturation (Frederiksen et al., 2021), whilst this appears not to occur (and not be necessary) in cones (Lobanova et al., 2010). The extent to which these processes underlie changes in human photoreceptor sensitivity post-bleach is not known. Future studies, combining tracking of recovery using different methods (psychophysical, electrophysiological, densitometric) in the same subjects following the same bleaches will be informative, as well as studies in patients in whom particular retinal pathways have been affected by pathogenic variants in particular genes. With current and future advances in genomic sequencing and its availability, such patients, although rare, are more likely to be identified, and these investigations are likely to shed light on both pathophysiological mechanisms in these conditions and the role of these pathways in normal physiology. Patients with defined genetic defects will also shed light on the precise origin of ERG components, enabling the ERG to be a more powerful tool for clinical and scientific investigation.

**Use of ERG tracking of dark adaptation in clinical contexts.** A number of studies have sought to employ ERG tracking of dark adaptation in clinical contexts, including the studies in age-related macular degeneration outlined above. Potential advantages of the ERG over psychophysical methods include its objective nature and less reliance on patient fixation and concentration, as well as direct assessment of responses at the level of

the retina. Several monogenic diseases affect the visual cycle, and, in some of these, quantitative assessment of retinal recovery following bleaches could contribute to genotype–phenotype correlation, identification of hypomorphic variants, evidence for pathogenicity or otherwise of genetic variants of uncertain significance, and potential outcome measures for trials of experimental therapies. Also, with the development of visual cycle modulators as a potential therapy for several conditions, efficacy in terms of retinoid cycle kinetics can be demonstrated and quantified, as discussed above in the case of emixustat.

Monogenic diseases in which there is known to be very slow rod-mediated dark adaptation include fundus albipunctatus and Oguchi disease. In both conditions, ISCEV standard dark-adapted ERGs are impaired whilst light-adapted ERGs are within normal limits; recovery of the dark-adapted ERG can sometimes be observed following overnight dark adaptation. Fundus albipunctatus is usually associated with bi-allelic pathogenic variants in *RDH5*, which encodes a retinoid cycle enzyme (Fig. 2). Oguchi disease is associated with bi-allelic pathogenic variants in *GRK1* or *SAG*. These genes respectively encode rhodopsin kinase and arrestin, both important in quenching the activity of activated rhodopsin. When these are defective, rhodopsin activity persists, leading to prolonged activation of phototransduction shutting off circulating current, leaving the rods electrically unresponsive to further flashes of light. With the profoundly slow rod dark adaptation in these conditions, monitoring ERGs dynamically in a single recording session using the methods described is not likely to be feasible (although it is possible that hypomorphic variants might exist in which this might be possible). Investigation of cone ERG recovery following bleaches (reflecting cone pigment regeneration) could be informative: it is possible that this will be impaired in fundus albipunctatus due to possible impaired delivery of retinoid to cones (depending on the extent to which the non-RPE cycle might compensate), whilst it might be expected to be intact in Oguchi disease. Patients with Sorsby fundus dystrophy (an autosomal dominant disorder associated with variants in the *TIMP3* gene) also usually have subjective dark adaptation impairment, demonstrable psychophysically. The thickening of Bruch's membrane in these patients might impair delivery of vitamin A from the bloodstream to the retinal pigment epithelium. The ERG methods described herein could provide an objective quantitative assessment of pan-retinal rod-mediated recovery in these patients.

When considering use of such testing protocols in patients and in clinical practice, the duration and potential discomfort of tests bear consideration. Delivering significant cone bleaches can involve intense exposures that might be uncomfortable, and use of flickering stimuli to track recovery can also be unpleasant. Measurement

of rod-mediated recovery, involving dimmer stimuli, is more comfortable for the subject; however, as recoveries are slower, the time burden is greater for the patient. Pupil dilatation, necessary for achieving consistent retinal illuminance, can add to discomfort, blurring vision for some hours afterwards. The size of visual electrophysiology equipment, and requirements for electrical isolation, can make testing within the space restrictions of many outpatient clinics more challenging. Portable, hand-held devices are now available: one device incorporates a camera so that the subject's eye can be monitored, as well as a pupilometer with automated adjustment of stimulus strength so that pharmacological mydriasis is not needed (described by Liu et al., 2018, and others). Development of shortened, refined testing protocols, using such devices might facilitate the incorporation of such tests into clinical practice in the future, and enable them to be more widely accessible.

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

## Additional information

### Competing interests

The authors have no competing interests.

### Author contributions

Conception and design, O.M.; literature search, analysis and synthesis of data, O.M. and X.J. Drafting manuscript, O.M. and X.J. Revisions of manuscript, O.M. and X.J. Both authors have read and approved the final version of this manuscript and agree to be accountable for all aspects of the work in ensuring that questions related to the accuracy or integrity of any part of the work are appropriately investigated and resolved. All persons designated as authors qualify for authorship, and all those who qualify for authorship are listed.

### Funding

This work was funded by the Wellcome Trust (206619/Z/17/Z) and Moorfields Eye Charity. The funding organisations had no role in the design or conduct of the research. Views expressed are those of the authors and not the funding organisations.

### Keywords

cone photoreceptors, dark adaptation, electroretinographyy, retina, retinal bipolar cells, rod photoreceptors

## Supporting information

Additional supporting information can be found online in the Supporting Information section at the end of the HTML view of the article. Supporting information files available:

**Peer Review History**

