## [Peer Review History · The Journal of Physiology]

Human retinal dark adaptation tracked in vivo with the electroretinogram: insights into processes underlying recovery of cone and rod-mediated vision

Xiaofan Jiang and Omar A R Mahroo

DOI: 10.1113/JP283105

Corresponding author(s): Omar Mahroo (oarm2@cam.ac.uk)

Review Timeline:

Submission Date:	27-Feb-2022
Editorial Decision:	03-Mar-2022
Resubmission Received:	15-Mar-2022
Editorial Decision:	08-Apr-2022
Revision Received:	17-Apr-2022
Accepted:	04-May-2022

Senior Editor: Laura Bennet

Reviewing Editor: William Taylor

Transaction Report:

Dear Professor Mahroo,

Re: JP-SR-2022-282071 "Human retinal dark adaptation tracked in vivo with the electroretinogram: insights into processes underlying recovery of cone and rod-mediated vision" by Xiaofan Jiang and Omar A Mahroo

Thank you for submitting your manuscript to The Journal of Physiology. It has been assessed by a Reviewing Editor and the report is are copied below.

Please let your co-authors know of the following editorial decision as quickly as possible.

As you will see, in its current form, the manuscript is not acceptable for publication in The Journal of Physiology. In comments to me, the Reviewing Editor expressed interest in the potential of this study, but much work still needs to be done (and this may include new experiments) in order to satisfactorily address the concerns raised in the reports.

In view of this interest, I would like to offer you the opportunity to carry out all of the changes requested in full, and to resubmit a new manuscript using the "Submit Special Case Resubmission for JP-SR-2022-282071..." on your homepage.

We cannot, of course, guarantee ultimate acceptance at this stage as the revisions required are substantial. However, we encourage you to consider the requested changes and resubmit your work to us if you are able to complete or address all changes.

A new manuscript would be renumbered and redated, but the original referees would be consulted wherever possible. An additional referee's opinion could be sought, if the Reviewing Editor felt it necessary. A full response to each of the reports should be uploaded with a new version.

I hope that the points raised in the reports will be helpful to you.

Yours sincerely,

Ian D. Forsythe
Deputy Editor-in-Chief
The Journal of Physiology
<https://jp.msubmit.net>
<http://jp.physoc.org>
The Physiological Society
Hodgkin Huxley House
30 Farringdon Lane
London, EC1R 3AW
UK
<http://www.physoc.org>
<http://journals.physoc.org>

EDITOR COMMENTS

Dear Authors, Looking through your article there seems to be multiple examples of raw data in the figures? We do not have the facility to ethically consider such data and their methods in a Review format. Please remove any unpublished data and adapt your figures. You may re-publish data with permission (and cite the source) in a Review but we particularly encourage use of diagrams and Biorender to generate diagrams that are informative and appeal to a wide audience. I look forward to receiving your revised MS.

ADDITIONAL FORMATTING REQUIREMENTS:

-Your MS must include a complete "Additional information section" with the following 4 headings and content:

Competing Interests: A statement regarding competing interests. If there are no competing interests, a statement to this effect must be included. All authors should disclose any conflict of interest in accordance with journal policy.

Author contributions: Each author should take responsibility for a particular section of the study and have contributed to writing the paper. Acquisition of funding, administrative support or the collection of data alone does not justify authorship; these contributions to the study should be listed in the Acknowledgements. Additional information such as 'X and Y have contributed equally to this work' may be added as a footnote on the title page.

It must be stated that all authors approved the final version of the manuscript and that all persons designated as authors qualify for authorship, and all those who qualify for authorship are listed.

Funding: Authors must indicate all sources of funding, including grant numbers. If authors have not received funding, this must be stated.

It is the responsibility of authors funded by RCUK to adhere to their policy regarding funding sources and underlying research material. The policy requires funding information to be included within the acknowledgement section of a paper. Guidance on how to acknowledge funding information is provided by the Research Information Network. The policy also requires all research papers, if applicable, to include a statement on how any underlying research materials, such as data, samples or models, can be accessed. However, the policy does not require that the data must be made open. If there are considered to be good or compelling reasons to protect access to the data, for example commercial confidentiality or legitimate sensitivities around data derived from potentially identifiable human participants, these should be included in the statement.

Acknowledgements: Acknowledgements should be the minimum consistent with courtesy. The wording of acknowledgements of scientific assistance or advice must have been seen and approved by the persons concerned. This section should not include details of funding.

-Please upload separate high quality figure files via the submission form.

-Author profile(s) must be uploaded via the submission form. Authors should submit a short biography (no more than 100 words for one author or 150 words in total for two authors) and a portrait photograph of the two leading authors on the paper. These should be uploaded, clearly labelled, with the manuscript submission. Any standard image format for the photograph is acceptable, but the resolution should be at least 300 dpi and preferably more. A group photograph of all authors is also acceptable, providing the biography for the whole group does not exceed 150 words.

JPHYSIOL Review: Response to Editor Comment

Editor comment

Dear Authors, Looking through your article there seems to be multiple examples of raw data in the figures? We do not have the facility to ethically consider such data and their methods in a Review format. Please remove any unpublished data and adapt your figures. You may re-publish data with permission (and cite the source) in a Review but we particularly encourage use of diagrams and Biorender to generate diagrams that are informative and appeal to a wide audience. I look forward to receiving your revised MS.

Response:

Thank you. We have adapted the figures to remove any raw or unpublished data, and updated the figure legends to emphasise that data are replotted from prior published studies where relevant.

Abstract Figure

We have revised this figure so that the row at the bottom contains schematic ERG traces and not traces from an actual subject (and made this clear in the legend).

Figure 1

The data here are replotted from previously published figures, and this has been made more clear in the legend.

Figure 3

We have revised this figure so that the right panel now contains a schematic ERG trace and not a trace from an actual subject.

Figure 4

We have revised this figure so that the panels now contain schematic ERG traces and not traces from an actual subject.

Figure 5

Panels A to D are all schematic. Only Panel E contains actual data – these data are replotted from data of Mahroo & Lamb (2012) – we have made this clear in the legend.

Figure 7

This is a comparison of ERG data and densitometry data – these are replotted from previous publications, and this has been made clear in the legend.

Dear Dr Mahroo,

Re: JP-SR-2022-283105X "Human retinal dark adaptation tracked in vivo with the electroretinogram: insights into processes underlying recovery of cone and rod-mediated vision" by Xiaofan Jiang and Omar A R Mahroo

Thank you for submitting your invited Review-Symposium to The Journal of Physiology. It has been assessed by a Reviewing Editor and by 2 expert referees and I am pleased to tell you that it is considered to be acceptable for publication following satisfactory revision.

The reports are copied at the end of this email. Please address all of the points and incorporate all requested revisions, or explain in your Response to Referees why a change has not been made.

NEW POLICY: In order to improve the transparency of its peer review process The Journal of Physiology publishes online as supporting information the peer review history of all articles accepted for publication. Readers will have access to decision letters, including all Editors' comments and referee reports, for each version of the manuscript and any author responses to peer review comments. Referees can decide whether or not they wish to be named on the peer review history document.

I hope you will find the comments helpful and have no difficulty in revising your manuscript within 4 weeks.

Your revised manuscript should be submitted online using the links in Author Tasks Link Not Available. This link is to the Corresponding Author's own account, if this will cause any problems when submitting the revised version please contact us.

The image files from the previous version are retained on the system. Please ensure you replace or remove any files that have been revised. Your revised submission should include:

- A Word file of the complete text (including figure legends any Tables);
- An Abstract Figure (with legend in the Article file)
- Each figure as a separate, high quality, file;
- A full Response to Referees;
- A copy of the manuscript with the changes highlighted.
- Author profile. A short biography (no more than 100 words for one author or 150 words in total for two authors) and a portrait photograph of the two leading authors on the paper. These should be uploaded, clearly labelled, with the manuscript submission. Any standard image format for the photograph is acceptable, but the resolution should be at least 300 dpi and preferably more.

- A 'Cover Art' file for consideration as the Issue's cover image;
- Appropriate Supporting Information (Video, audio or data set https://jp.msubmit.net/cgi-bin/main.plex?form_type=display_requirements#supp).

To create your 'Response to Referees' copy all the reports, including any comments from the Reviewing Editor into a Word, or similar, file and respond to each point in colour or CAPITALS and upload this when you submit your revision.

I look forward to receiving your revised submission.

If you have any queries please reply to this email and staff will be happy to assist.

Yours sincerely,

Ian D. Forsythe
Deputy Editor-in-Chief
The Journal of Physiology
<https://jp.msubmit.net>
<http://jp.physoc.org>
The Physiological Society
Hodgkin Huxley House
30 Farringdon Lane
London, EC1R 3AW
UK
<http://www.physoc.org>
<http://journals.physoc.org>

EDITOR COMMENTS

Reviewing Editor:

The referees have some comments and suggestions, which the authors should find helpful in revision.

Some additional suggestions:

In the abstract and the first paragraph of the concluding remarks section, the authors note a critical difference in the recovery of the dark-current in rods and cones - the latter are insensitive to the presence of unbound opsin. Are there studies that could be cited that point to a potential molecular basis for this difference?

Abstract figure:

- a. Is it possible to temporally align the rod-bwave schematics at the bottom with the recovery panels at the top?
- b. It would be informative to have an approximate time-calibration.
- c. Is it possible to shuffle the elements so that the text in the yellow box on the left can be displayed horizontally?

P12: "For standard clinical testing...". Would it be useful to document the light intensities used under these standard conditions?

Fig. 4: The top panel looks like a photo of a figure. The coloured backgrounds are distracting, at least to me, and don't add information. Suggest replotting without backgrounds. Consider moving the vertical "Dark-adapted" and "Light-adapted" labels above the respective panels oriented horizontally.

P16: section entitled "Cone-driven a-wave amplitudes indicate rapid recovery of circulating current, with slower, rate-limited recovery of photopigment"; the authors note that "These were bright and dim flashes ... to probe recovery of cone circulating current and cone photopigment respectively". Perhaps this is obvious to those in the field, but for a non-expert reader, it would help to explain a little more how recovery of the circulating current can be distinguished from photopigment recovery in ERG recordings.

REFEREE COMMENTS

Referee #1:

This is a very clearly written and comprehensive review of the use of ERGs to monitor dark adaptation of human rod and cone photoreceptors following bleaching exposures. Coverage extends from the basic cellular mechanisms, through the responses of rods and cones in normal human subjects, to the application of the techniques in the clinic, not only to study disease mechanisms but also for future more widespread use with hand-held devices and slimmed-down protocols. The diagrams and the text explanations are set out in a manner that is easy to follow, yet that deals with the complications and difficulties of the methods. I support publication enthusiastically, and I have only minor suggestions for improvement, as follows.

Page 6, first line of Introduction: Replace "several" with something more explicit?

Next paragraph: "retinal response of the retina". Delete first "retinal"?

Page 6, final paragraph: Perhaps say "following cessation of bright light exposures" or similar.

Page 8, paragraph 2: "a particular" occurs twice in close succession. Replace the first?

Same line: "Interestingly the time taken". Could this be indicated in Fig. 1B with arrow(s)?

Page 9, first full paragraph: It would help to indicate that the all-trans retinal has been released prior to conversion to retinal.

Line 3: "to form rhodopsin once more". Avoid giving the reader the impression that rhodopsin doesn't re-form until the retinoid has been right round the cycle.

Line 8: 'a "cone-specific" visual cycle'. It might be worth mentioning the light-driven cycle involving RGR reported from the laboratory of G.H. Travis, while emphasizing that this review covers recovery in darkness.

Final paragraph: "Rushton et al. (1955)". Rushton & Henry (1955)?

Page 10, first paragraph. For historical context, it might be worth mentioning the ERG experiments of Granit (1933) J. Physiol., even though he did not study dark adaptation.

Final paragraph: "reducing glutamate release". Would it be worth mentioning the high rate of steady release in darkness? And also the specialized ribbon synapses?

Page 23: "varying intensity". Avoid suggesting that each intensity is not constant.

Page 25: "following a 2 in bleaching exposure". "2 min".

Referee #2:

The paper reviews the physiological processes involved in recovery of visual sensitivity during dark adaptation in rod and cone photoreceptors, with a focus on the role that ERGs can play in studying the processes, and in evaluating therapeutic interventions in certain retinal diseases. This review from the lab of an expert in the field is well presented and informative.

I only have some minor comments

1. Abstract figure legend: Please explain the bleach recovery images at the top of the figure - these images will be new to many readers.

2. Pg. 15, wording, "groups according by" (accordingly?)

3. Pg. 25, Photopic flicker paragraph, middle, " in following a 2 in bleaching exposure" (2 min?)

4. Pg. 35, Table 1: Cameron et al, second column, "comparison of with responses on different" ??

END OF COMMENTS

1st Confidential Review

15-Mar-2022

JP-SR-2022-283105X

Human retinal dark adaptation tracked in vivo with the electroretinogram: insights into processes underlying recovery of cone and rod-mediated vision

Response to Reviewers

We thank the editor and reviewers for their thoughtful and helpful comments. We have amended the manuscript accordingly, and we include our responses in bold below.

EDITOR COMMENTS

Reviewing Editor:

The referees have some comments and suggestions, which the authors should find helpful in revision.

Some additional suggestions:

In the abstract and the first paragraph of the concluding remarks section, the authors note a critical difference in the recovery of the dark-current in rods and cones - the latter are insensitive to the presence of unbound opsin. Are there studies that could be cited that point to a potential molecular basis for this difference?

The precise molecular basis is not known, but the following text has now been added in the concluding remarks section with relevant references:

“Whilst bleached photopigment has been shown to activate transduction in both rods and cones (Cornwall et al., 1995; Matthews et al., 1996), the molecular basis for the difference in effect on circulating current following a bleach remains to be fully elucidated. Single-cell recordings have shown that mouse M-cones can maintain circulating current near the dark-adapted level even with more than 90% of opsin in the bleached state (Nikonov et al., 2006), consistent with findings from the human ERG studies described above (Kenkre et al., 2005). Cone phototransduction has lower amplification than rods, with shorter lifetimes of active photopigment, and faster inactivation of activated transducin (Nikonov et al., 2006, Lobanova et al., 2010). The lower overall gain in the transduction cascade in cones might play a role in allowing maintenance of circulating current similar to dark levels even in the presence of high levels of opsin post-bleach.”

Abstract figure:

a. Is it possible to temporally align the rod-bwave schematics at the bottom with the recovery panels at the top?

The recovery panels at the top are illustrative and not meant to be accurate descriptions of vision at even time intervals. Thus, we have deliberately avoided making an alignment with the rod b-wave schematics at the bottom of the figure to avoid misleading the reader. To make this more clear, the following has been added to the beginning of the abstract figure legend:

“The upper panels represent (illustratively rather than accurately) recovery in visual sensitivity over time following a bright light exposure (“bleach”).”

b. It would be informative to have an approximate time-calibration.

This has now been added.

c. Is it possible to shuffle the elements so that the text in the yellow box on the left can be displayed horizontally?

This has been amended as suggested.

P12: "For standard clinical testing...". Would it be useful to document the light intensities used under these standard conditions?

Figure 4 gives the standard light intensities. This has now been stated explicitly in the text.

The final sentence of the preceding paragraph now states:

“Figure 4 illustrates schematically the form of responses recorded from a healthy individual to standard clinical stimuli (standard stimulus strengths are given in the legend).”

And the first sentence of the paragraph alluded to now details the strength of the standard light-adapting background, and reads as follows:

“For standard clinical testing, retinal responses are thus interrogated at one of two steady states, namely (1) dark-adapted, or (2) light-adapted to a specific background luminance (namely a white background of 30 cd m⁻²).”

Fig. 4: The top panel looks like a photo of a figure. The coloured backgrounds are distracting, at least to me, and don't add information. Suggest replotting without backgrounds. Consider moving the vertical "Dark-adapted" and "Light-adapted" labels above the respective panels oriented horizontally.

The figure has been revised as suggested.

P16: section entitled "Cone-driven a-wave amplitudes indicate rapid recovery of circulating current, with slower, rate-limited recovery of photopigment"; the authors note that "These were bright and dim flashes ... to probe recovery of cone circulating current and cone photopigment respectively". Perhaps this is obvious to those in the field, but for a non-expert reader, it would help to explain a little more how recovery of the circulating current can be distinguished from photopigment recovery in ERG recordings.

Essentially, bright-flash response-amplitudes are taken to reflect circulating current, as these represent the response to a flash sufficiently bright to shut off (transiently) all of the circulating current. For calculation of photopigment (usually from dim-flash responses), it is assumed the circulating current has fully recovered (as suggested by the bright-flash results), and a response-intensity relation is used to derive the "effective intensity" of post-bleach flashes (as shown in Figure 5). The text has been modified to provide more explanation.

The discussion of bright-flash responses now reads (relevant changes highlighter in red):

"The authors delivered flashes on the rod-saturating background before and after intense bleaching exposures. These were bright and dim flashes (measuring the a-wave at fixed post-flash times in both cases) to probe recovery of cone circulating current and cone photopigment respectively. **Based on the electrical responses of single photoreceptors, the amplitude of the response to a bright flash (that transiently but fully shuts off the photoreceptor circulating current) will give an estimate of the circulating current. The basis of the use of the dim-flash a-wave to estimate photopigment will be discussed subsequently.** The bright-flash a-wave amplitude recovered within 30 seconds following an almost total bleach, and within a few seconds following a partial bleach, indicating a recovery of cone circulating current that is orders of magnitude faster than in rods following similar bleaching exposures."

The later discussion of dim-flash responses now reads (relevant changes highlighter in red):

"In the study of Paupoo et al. (2000), the dim-flash cone-driven a-wave recovered over several minutes, consistent with this reflecting the kinetics of cone photopigment regeneration. **Based on the analysis of Lamb & Pugh (1992), the response to a dim flash, measured at early post-flash times, is expected to be proportional to the product of three factors, namely the effective flash strength (which, for a constant-strength flash, will in fact be proportional to photopigment levels), the amplification constant of (the activation phase of) phototransduction, and the circulating current (Lamb & Pugh, 1992). If cone circulating current recovers very quickly (as shown from bright-flash responses), and if the amplification constant is assumed to be unchanged (as appeared to be the case for rods from the investigations of Thomas & Lamb, 1999), then the response-amplitude would**

reflect photopigment levels; this was the basis of the conclusions of Paupoo et al. (2000) in relation to photopigment recovery.

However, the flash strengths used by Paupoo et al (2000), though dim, are outside the linear range, such that response-amplitude as measured is not proportional to flash strength, and therefore will not be proportional to pigment levels. Direct modelling of photopigment kinetics requires accounting for the non-linearity in the response-intensity relation. Specific adjustment for this non-linearity was performed by Mahroo & Lamb (2004).”

REFeree COMMENTS

Referee #1:

This is a very clearly written and comprehensive review of the use of ERGs to monitor dark adaptation of human rod and cone photoreceptors following bleaching exposures. Coverage extends from the basic cellular mechanisms, through the responses of rods and cones in normal human subjects, to the application of the techniques in the clinic, not only to study disease mechanisms but also for future more widespread use with hand-held devices and slimmed-down protocols. The diagrams and the text explanations are set out in a manner that is easy to follow, yet that deals with the complications and difficulties of the methods. I support publication enthusiastically, and I have only minor suggestions for improvement, as follows.

Page 6, first line of Introduction: Replace "several" with something more explicit?

This has been amended and now reads:

“The visual system adapts by appropriately adjusting sensitivity over a remarkable range of background intensities spanning more than nine log units.”

Next paragraph: "retinal response of the retina". Delete first "retinal"?

Thanks for spotting this. The first “retinal” has been deleted.

Page 6, final paragraph: Perhaps say "following cessation of bright light exposures" or similar.

This has now been added to the subheading.

Page 8, paragraph 2: "a particular" occurs twice in close succession. Replace the first?

The first "particular" has now been replaced with "specific".

Same line: "Interestingly the time taken". Could this be indicated in Fig. 1B with arrow(s)?

A horizontal line has been added to indicate an example threshold intensity, and the text has been modified accordingly. The following has been added to the figure legend:

"The horizontal broken red line denotes an arbitrary level of threshold elevation within this region; the red circles denote the points at which recoveries cross this line. The time taken to such a level of threshold elevation (denoted by the x-axis value for each red circle) was shown by Lamb (1981) to be linearly related to the initial bleach level, for large bleaches, suggesting removal of a bleaching photoproduct at a constant linear rate."

Page 9, first full paragraph: It would help to indicate that the all-trans retinal has been released prior to conversion to retinal.

Some studies appear to indicate that the all-trans retinal is converted to all-trans retinol whilst still attached to the opsin, so we have amended the text as follows:

"The all-trans retinal is released from opsin and is converted to all-trans retinol (possibly prior to release)."

Line 3: "to form rhodopsin once more". Avoid giving the reader the impression that rhodopsin doesn't re-form until the retinoid has been right round the cycle.

The "once more" has now been removed.

Line 8: 'a "cone-specific" visual cycle'. It might be worth mentioning the light-driven cycle involving RGR reported from the laboratory of G.H. Travis, while emphasizing that this review covers recovery in darkness.

The following has now been added to this paragraph:

"The retinal G-protein-coupled receptor (RGR, not shown in Figure 2), provides further contributions to retinoid recycling (Radu *et al.*, 2008; Morshedian *et al.*, 2019). RGR is expressed in the RPE and in Muller cells, and mediates light-driven processes, likely to be

of less relevance to recovery in the dark, which is the focus of this review.”

Final paragraph: "Rushton et al. (1955)". Rushton & Henry (1955)?

This has now been amended.

Page 10, first paragraph. For historical context, it might be worth mentioning the ERG experiments of Granit (1933) *J. Physiol.*, even though he did not study dark adaptation.

This historical context has now been given, with the start of the paragraph now reading as follows:

“Studies describing ERG recordings *in vivo* were published in the 19th century, and the later experiments of Granit (1933) in the decerebrate cat were important in describing separable components in the ERG, though he did not specifically investigate dark adaptation. Electrophysiology was later used specifically to explore recovery of visual system responses following bleaching exposures, with the earliest studies preceding the first retinal densitometry studies, and demonstrating that the reduction in visual system scotopic sensitivity was clearly detectable at the level of the retina.”

Final paragraph: "reducing glutamate release". Would it be worth mentioning the high rate of steady release in darkness? And also the specialized ribbon synapses?

This has now been mentioned, with the following added to the start of the paragraph:

“The photoreceptor circulating current flowing in the dark depolarises the cell, leading to a steady release of glutamate at the synaptic terminal. The photoreceptor to bipolar cell synapse (reviewed recently by Burger *et al.*, 2021) is a ribbon synapse, characterised ultrastructurally by the presence of dense regions of aligned synaptic vesicles.”

Page 23: "varying intensity". Avoid suggesting that each intensity is not constant.

This phrase has been changed to “different intensities”.

Page 25: "following a 2 in bleaching exposure". "2 min".

This has now been corrected.

Referee #2:

The paper reviews the physiological processes involved in recovery of visual sensitivity during dark adaptation in rod and cone photoreceptors, with a focus on the role that ERGs can play in studying the processes, and in evaluating therapeutic interventions in certain retinal diseases. This review from the lab of an expert in the field is well presented and informative.

I only have some minor comments

1. Abstract figure legend: Please explain the bleach recovery images at the top of the figure - these images will be new to many readers.

These are meant to symbolically show recovery in vision following a bleach rather than be accurate representations of visual sensitivity. To make this clearer, the following has been added to the abstract figure legend:

“The upper panels represent (illustratively rather than accurately) recovery in visual sensitivity over time following a bright light exposure (“bleach”).”

2. Pg. 15, wording, "groups according by" (accordingly?)

This has been corrected to “according to”.

3. Pg. 25, Photopic flicker paragraph, middle, " in following a 2 in bleaching exposure" (2 min?)

This has now been corrected.

4. Pg. 35, Table 1: Cameron et al, second column, "comparison of with responses on different" ??

This has been corrected, and now reads:

“Comparison with responses on different backgrounds to derive estimates of equivalent backgrounds.”

END OF COMMENTS

Dear Professor Mahroo,

Re: JP-SR-2022-283105XR1 "Human retinal dark adaptation tracked in vivo with the electroretinogram: insights into processes underlying recovery of cone and rod-mediated vision" by Xiaofan Jiang
Omar A R Mahroo

I am pleased to tell you that your Symposium Review article has been accepted for publication in The Journal of Physiology, subject to any modifications to the text that may be required by the Journal Office to conform to House rules.

NEW POLICY: In order to improve the transparency of its peer review process The Journal of Physiology publishes online as supporting information the peer review history of all articles accepted for publication. Readers will have access to decision letters, including all Editors' comments and referee reports, for each version of the manuscript and any author responses to peer review comments. Referees can decide whether or not they wish to be named on the peer review history document.

The last Word version of the paper submitted will be used by the Production Editors to prepare your proof. When this is ready you will receive an email containing a link to Wiley's Online Proofing System. The proof should be checked and corrected as quickly as possible.

All queries at proof stage should be sent to tjp@wiley.com

The accepted version of the manuscript is the version that will be published online until the copy edited and typeset version is available. Authors should note that it is too late at this point to offer corrections prior to proofing. Major corrections at proof stage, such as changes to figures, will be referred to the Reviewing Editor for approval before they can be incorporated. Only minor changes, such as to style and consistency, should be made a proof stage. Changes that need to be made after proof stage will usually require a formal correction notice.

Are you on Twitter? Once your paper is online, why not share your achievement with your followers. Please tag The Journal (@jphysiol) in any tweets and we will share your accepted paper with our 22,000+ followers!

Yours sincerely,

Professor Laura Bennet
Senior Editor
The Journal of Physiology
<https://jp.msubmit.net>
<http://jp.physoc.org>
The Physiological Society
Hodgkin Huxley House
30 Farringdon Lane
London, EC1R 3AW
UK
<http://www.physoc.org>
<http://journals.physoc.org>

Comments:

Reviewing Editor:
Nice review!

REFeree COMMENTS:

Referee #3:

The authors have fully addressed all of my previous comments and suggestions, and the revised paper reads very nicely and represents a comprehensive review of the field.

Referee #4:

This review is a useful contribution that clearly describes human retinal dark adaptation and the use of the electroretinogram to noninvasively study it.

*** IMPORTANT NOTICE ABOUT OPEN ACCESS ***

Information about Open Access policies can be found here <https://physoc.onlinelibrary.wiley.com/hub/access-policies>

To assist authors whose funding agencies mandate public access to published research findings sooner than 12 months after publication The Journal of Physiology allows authors to pay an open access (OA) fee to have their papers made freely available immediately on publication.

You will receive an email from Wiley with details on how to register or log-in to Wiley Authors Services where you will be able to place an OnlineOpen order.

You can check if your funder or institution has a Wiley Open Access Account here <https://authorservices.wiley.com/author-resources/Journal-Authors/licensing-and-open-access/open-access/author-compliance-tool.html>

Your article will be made Open Access upon publication, or as soon as payment is received.

If you wish to put your paper on an OA website such as PMC or UKPMC or your institutional repository within 12 months of publication you must pay the open access fee, which covers the cost of publication.

OnlineOpen articles are deposited in PubMed Central (PMC) and PMC mirror sites. Authors of OnlineOpen articles are permitted to post the final, published PDF of their article on a website, institutional repository, or other free public server, immediately on publication.

Note to NIH-funded authors: The Journal of Physiology is published on PMC 12 months after publication, NIH-funded authors DO NOT NEED to pay to publish and DO NOT NEED to post their accepted papers on PMC.